# Closed-loop brain stimulation augments fear extinction in male rats

Rodrigo Ordoñez Sierra[1,10], Lizeth Katherine Pedraza[1,10], Lívia Barcsai [1,2,3], Andrea Pejin[1,2,3], Qun Li [1], Gábor Kozák [1], Yuichi Takeuchi [1,4], Anett J. Nagy[1,2,3], Magor L. Lőrincz [1,5,6], Orrin Devinsky [7], György Buzsáki [8,9] & Antal Berényi [1,2,3,8] ✉

Dysregulated fear reactions can result from maladaptive processing of trauma-related memories. In post-traumatic stress disorder (PTSD) and other psychiatric disorders, dysfunctional extinction learning prevents discretization of trauma-related memory engrams and generalizes fear responses. Although PTSD may be viewed as a memory-based disorder, no approved treatments target pathological fear memory processing. Hippocampal sharp wave-ripples (SWRs) and concurrent neocortical oscillations are scaffolds to consolidate contextual memory, but their role during fear processing remains poorly understood. Here, we show that closed-loop, SWR triggered neuromodulation of the medial forebrain bundle (MFB) can enhance fear extinction consolidation in male rats. The modified fear memories became resistant to induced recall (i.e., 'renewal' and 'reinstatement') and did not reemerge spontaneously. These effects were mediated by D2 receptor signaling-induced synaptic remodeling in the basolateral amygdala. Our results demonstrate that SWR-triggered closed-loop stimulation of the MFB reward system enhances extinction of fearful memories and reducing fear expression across different contexts and preventing excessive and persistent fear responses. These findings highlight the potential of neuromodulation to augment extinction learning and provide a new avenue to develop treatments for anxiety disorders.

Learning unpleasant things and remembering them is advantageous for the organism for avoiding future reoccurrences. Memories that are irrelevant to survival or adaptation tend to fade away either by graceful degradation[1,2] or by another type of learning called active extinction[3,4]. Extinction learning, the process of reducing the expression of learned fear responses, is essential for adaptive behavior in response to traumatic experiences.

However, in some pathological scenarios, extinction learning is often impaired, leading to persistent and maladaptive fear responses[5]. For example, post-traumatic stress disorder (PTSD) is a debilitating psychiatric disorder resulting from direct or indirect exposure to stressful events, threats, or life-threatening events perceived to compromise personal physical or mental safety[6–8]. Symptoms include intense feelings of unprovoked fear, panic attacks, anxiety; intrusive

[1]MTA-SZTE 'Momentum' Oscillatory Neuronal Networks Research Group, Department of Physiology, University of Szeged, Szeged 6720, Hungary. [2]HCEMM-SZTE Magnetotherapeutics Research Group, University of Szeged, Szeged 6720, Hungary. [3]Neunos Inc, Boston, MA 02108, USA. [4]Department of Biopharmaceutical Sciences and Pharmacy, Faculty of Pharmaceutical Sciences, Hokkaido University, Sapporo, Japan. [5]Department of Physiology, Anatomy and Neuroscience, Faculty of Sciences University of Szeged, Szeged 6726, Hungary. [6]Neuroscience Division, Cardiff University, Museum Avenue, Cardiff CF10 3AX, UK. [7]Department of Neurology, NYU Langone Comprehensive Epilepsy Center, NYU Grossman School of Medicine, New York, NY 10016, USA. [8]Neuroscience Institute, New York University, New York, NY 10016, USA. [9]Center for Neural Science, New York University, New York, NY 10016, USA. [10]These authors contributed equally: Rodrigo Ordoñez Sierra, Lizeth Katherine Pedraza. ✉e-mail: drberenyi@gmail.com

fear memories during wakefulness or in nightmares, fear generalization, and avoiding similar but neutral stimuli[9,10]. PTSD is highly resistant to psycho- and pharmacotherapy[11–13].

Previous studies have demonstrated that hippocampal sharp-wave ripples (SWRs) play a critical role in the consolidation of fear memories[14], and that closed-loop stimulation of the reward system can enhance memory consolidation[15]. Exposure-based extinction procedures have been found to reduce fear in a context-dependent manner, suggesting that the hippocampal representation of the extinction context drives fear attenuation[16]. The activity in the basolateral amygdala decreases when conditioning stimuli (CS+) are presented in the same context used for extinction but increases following non-extinction exposure to the CS+[17]. Furthermore, inactivation of the hippocampus has been found to enhance extinction to the CS+ and promote low fear expression in environments different from the extinction context[18,19].

Excitatory neurons in the basolateral amygdala have been shown to respond to both reward and punishment and have been proposed to be involved in mediating reward signaling induced by the omission of an unconditioned stimulus during extinction[20]. Additionally, these neurons participate in a mutual inhibition process[21]. Based on these findings, we hypothesize that manipulating internal reward signals during extinction learning could facilitate the extinction of memories, thereby reducing excessive fear reactions in inappropriate contexts.

Here, we explore whether SWR-triggered stimulation of the reward system through medial-forebrain bundle (MFB) can augment extinction learning. Our findings suggest that SWRs are crucial for mediating fear extinction, and that closed-loop neuromodulation targeting oscillatory activity related to memory processing could be a promising intervention for reducing excessive fear reactions in inappropriate contexts. Specifically, our experiments demonstrate that selective suppression of SWRs after extinction delayed fear attenuation, indicating that intact SWRs are necessary for extinction learning. Furthermore, our results show that SWR-triggered closed-loop stimulation of the reward system through MFB enhances the extinction of fearful memories, resulting in reduced fear expression across different contexts and preventing excessive and persistent fear responses. Overall, our study suggests that rewarding brain stimulation may be a promising approach to augment extinction learning, potentially beneficial to alleviate PTSD symptoms.

## Results

### SWR-driven closed-loop electrical stimulation of the medial-forebrain bundle accelerates extinction and prevents fear recovery

Rats were subjected to a single session of fear conditioning (5 pairings of conditions stimulus, CS+ and unconditioned stimulus, US, i.e., footshock using 1 mA current) to develop PTSD-like phenotypes (Supplementary Fig. 3a–e) followed by fear extinction training over multiple days (twenty re-exposures/day in four blocks to CS+ in a novel context without US) until a remission criterion (reduction of freezing behavior to <20% of the initial freezing) was reached or up to maximum seven days (Fig. 1a). During the extinction protocol, one group of rats received closed-loop stimulation of the MFB during hippocampal sharp-wave ripples (SWRs) (fourteen 1-ms long, 100 μA square-wave pulses at 140 Hz; Fig. 1b) to assign a reward signal to the replayed extinction memory, another group received jittered stimulation (open-loop), and a control group received no stimulation (Fig. 1c). The experimental protocol involved conducting stimulation and recording sessions for a duration of one hour immediately following the extinction procedure. To minimize the influence of novelty, the stimulation was carried out within the animals' home cage. Fear-related behavioral performance was tested using different tests to assess the persistence of the extinction memory as follows. Animals were exposed to CS+ in a hybrid context mixing new features with the conditioning context

following extinction ('RENEWAL TEST') and by unpredictable exposure to the US ('REINSTATEMENT TEST'). The persistence of the extinction was assessed by exposing the animals to CS+25 days following extinction ('REMOTE TEST').

We found that the average online detection rate of SWRs was 80.38 ± 1.349% compared to the post hoc detection rate. False positive detection rate was 7.750 ± 1.830%, while the rate of missed detections was 11.88 ± 7.67% (Fig. 1d), which further confirms the high accuracy of our detection method. The minimum delay for triggering the stimulation after SWR detection was 15 ms, while the maximum was 27 ms. Notably, the majority of SWR events were detected between 18 and 21 ms before the onset of stimulation (Fig. 1e; Supplementary Fig. 1), highlighting the precise timing of our closed-loop stimulation approach.

Our results demonstrated that the global architecture of sleep and the distinct sleep stages were not affected by the closed-loop neuromodulation of MFB (Fig. 1f; Supplementary Fig. 2), suggesting the observed effects on fear memories were specific to the closed-loop stimulation and not a result of changes in sleep patterns. The rewarding properties of the MFB stimulation were verified using a conditioned place preference task (Supplementary Fig. 3f). No significant differences were found in the fear expression between groups in the test after conditioning to CS+ (Fig. 1g), contextual fear conditioning (Supplementary Fig. 5) or after the first or the last extinction days (Fig. 1h). Supplementary Data 1 shows the results of descriptive and comparative statistics.

While our findings demonstrated that extinction can lead to the overcoming of fear (as evidenced by individual extinction rates shown in Supplementary Fig. 6), animals that were exposed to closed-loop stimulation required fewer extinction sessions to achieve the remission criterion of <20% initial freezing compared to the open-loop and non-stimulated groups (Fig. 1i) suggesting that closed-loop neuromodulation of MFB can enhance the effectiveness of fear extinction.

Following the exposure to the 'renewal test' in a hybrid context there was a significant decrease in fear expression in the closed-loop treated animals compared to the open-loop and non-stimulated groups (Fig. 1j). These results indicate that closed-loop MFB stimulation during SWRs can enhance fear extinction, decrease the time needed to achieve fear attenuation and maintain freezing levels low in challenging situations such as exposure to hybrid contexts resembling the learning contingencies.

To assess the persistence of the effects, animals were exposed to a 'remote test' 25 days following the renewal in the hybrid context. Animals were kept in their home cages between the renewal and remote tests. Freezing in closed-loop stimulated animals remained at low levels compared to the open-loop and non-stimulated group (Fig. 1k), suggesting fear attenuation induced by closed-loop MFB stimulation was resistant to spontaneous recovery and persisted over time.

Finally, we quantified Δ freezing as reduced fear reactions between those after fear condition and the remote test (Δ freezing = Freezing extinction − Freezing test CS+) to reveal the overall effect of the interventions (Supplementary Fig. 4 shows the performance of individual animals in each group). Closed-loop simulated animals had stronger fear reduction than open-loop and non-stimulated animals (Fig. 1l). Together, closed-loop neuromodulation of the reward system triggered by memory consolidation-related neuronal oscillations accelerates fear extinction and promotes persistent low fear expression.

### Exploring the contribution of extinction learning and potential side effects during closed-loop MFB stimulation

We investigated whether closed-loop MFB stimulation without any extinction training could reduce fear, as MFB stimulation is known to be rewarding. To test this, after fear conditioning with 5 pairings of CS

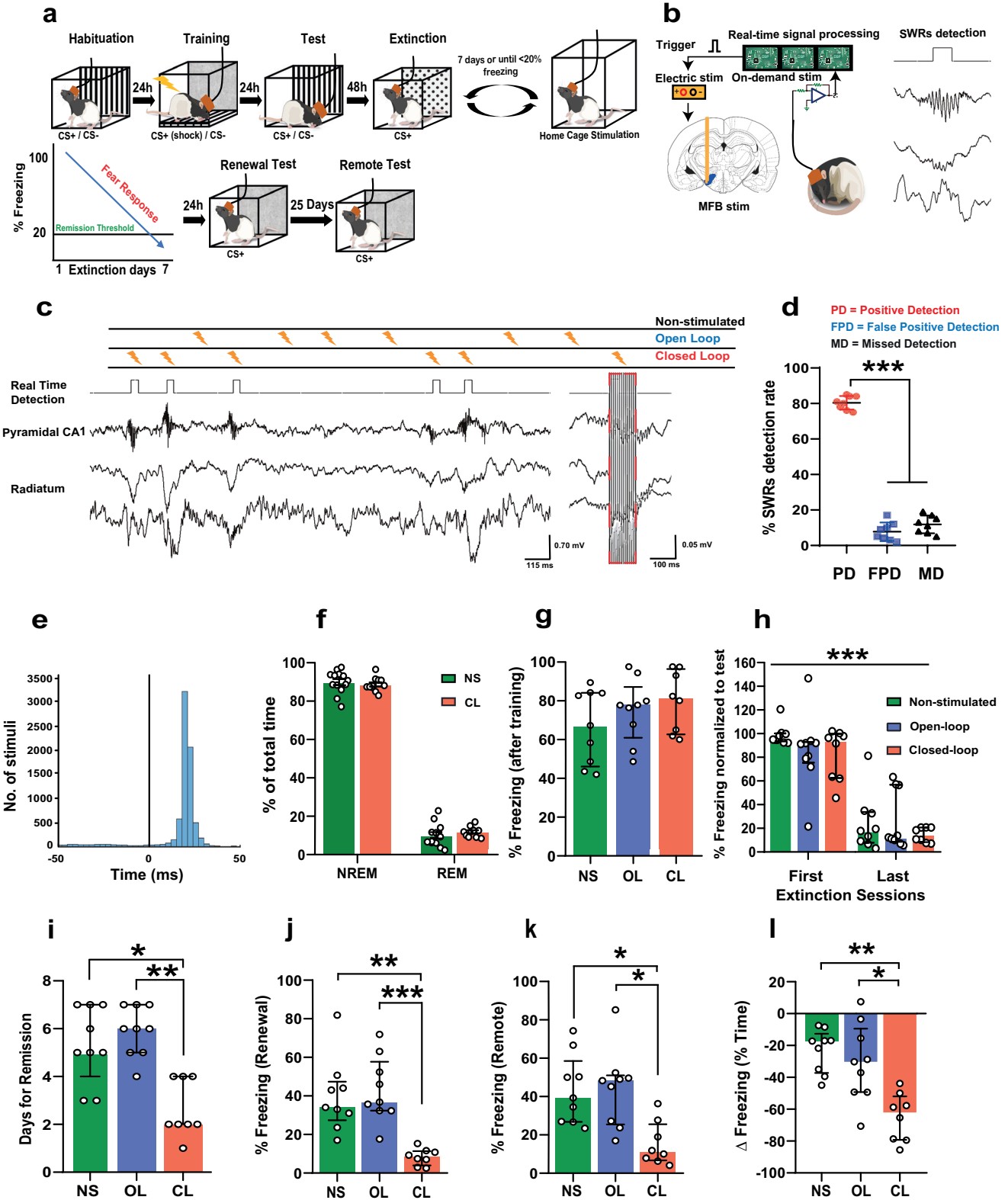

+US (1 mA), animals received SWR-triggered closed-loop stimulation during sleep for three consecutive days but were not exposed to the extinction paradigm (Fig. 2a).

To match the mean number of extinction sessions required for closed-loop animals to achieve the remission criterion (Fig. 1i), the number of stimulation sessions was set to 3 days, and the stimulation durations were kept the same as in the previous experiment with

extinction. The non-stimulated (NS) control group underwent identical fear conditioning and spent three days in their home cage without any intervention. No significant differences were observed between the two groups immediately after CS+ conditioning (Fig. 2b) or after three days of stimulation sessions (Fig. 2c). Thus, the closed-loop SWR-triggered stimulation alone, without extinction, did not lead to a decrease in fear expression.

**Fig. 1 | Closed-loop SWR-timed medial-forebrain bundle electrical stimulation attenuates fear memories. a** Schematics of the experimental design. **b** A custom threshold crossing algorithm was used to trigger the MFB stimulation following online detections of SWRs. **c** Closed- loop stimulation consisted of MFB stimulation during the detected SWR events, open-loop stimulation was similar to closed-loop but stimulation was jittered from SWRs (top). Representative LFP signals from dorsal hippocampus showing SWR events and stimulation pattern (right). **d** Average online detection rate of SWR events (PD Positive detection; FPD False positive detection, MD Missed detection; $n = 8$) (one-way ANOVA: $F_{(2,21)} = 602.1$, $P < 0.0001$). **e** Delay of stimulation triggering from the beginning of the SWRs. The largest number of stimuli (blue peak) were delivered between 18 and 21 ms after the SWR onset (black line: time zero). **f** No difference in sleep architecture between closed-loop (CL) and non-stimulated (NS) animals during non-REM (NREM) sleep (Unpaired $t$ test: $t_{(22)} = 0.5977$, $P = 0.5561$, two-tailed) and REM sleep (Unpaired $t$ test: $t_{(22)} = 0.9459$, $P = 0.3545$, two-tailed). Data represent mean ± SEM (Number of sessions: (NS) $n = 13$; (CL) $n = 11$). **g** No difference in fear expression in response to the CS+ following training between the three experimental groups (Kruskal–Wallis test: $H = 1.737$, $P = 0.4195$) (non-stimulated (NS) $n = 9$; open-loop (OL) $n = 9$; closed-

loop (CL) $n = 8$). **h** No difference between the fear expression of the three groups during the first 5 CS+ block after first and last extinction days. There was a significant decrease in fear expression over time (mixed ANOVA: $F_{(1,23)} = 164.2$, $P < 0.0001$, time factor). Values are normalized to the freezing expressed immediately after footshock training (i.e., "Test"). **i** Animals exposed to closed-loop stimulation required less extinction sessions to achieve the remission criterion compared to the open-loop and non-stimulated groups (Kruskal–Wallis test: $H = 13.60$, $P = 0.0011$). **j** Closed-loop neuromodulation-induced lower fear expression during the renewal test in a hybrid context (Kruskal–Wallis test: $H = 16.21$, $P = 0.0003$). **k** Closed-loop neuromodulation prevented spontaneous fear recovery 25 days after extinction (Kruskal–Wallis test: $H = 10.38$, $P = 0.0056$). **l** Closed-loop neuromodulation produces the greatest reduction in fear between post-training and remote testing following extinction (Kruskal–Wallis test: $H = 13.06$, $P = 0.0015$). *$P < 0.05$, **$P < 0.01$, ***$P < 0.001$. Bar plots and error bars represent medians and interquartile ranges, individual data points are also displayed. Detailed statistics are shown in Supplementary Data 1. Source data provided as a Source Data file. Silhouettes on **a**, **b** are obtained from https://github.com/eackermann/ratpack under MIT License.

We next tested if the SWR-triggered closed-loop stimulation interferes with already consolidated non-fear-related memories as a non-specific detrimental effect. For this purpose, the animals were trained in a spatial memory task, in which a randomly alternated visual cue indicated the correct choice in a T-maze to receive a reward (frootloops pellet). They underwent a total of 20 trials per day until achieving 80% of correct choice. After completing the spatial memory task, the animals underwent fear conditioning, extinction, and stimulation sessions in the same way as in the previous experiment until achieving remission (Fig. 2d). During the extinction procedure, the animals were also retested in the same spatial memory task each day, with a five-hour gap between the extinction + stimulation sessions and T-maze task. The order of the behavioral tasks was randomized across the experiment. Both OL and CL stimulated animals maintained performance in the T-maze task (Fig. 2e). The individual performance of animals during the fear conditioning and extinction procedure is shown in Fig. 2f, g. Moreover, the extinction enhancement induced by CL neuromodulation was preserved (Supplementary Fig. 8). These results suggest that closed-loop stimulation alone is not sufficient to reduce fear expression and must be coupled with extinction learning. Moreover, already consolidated spatial memories are not affected by the stimulation.

## SWRs are required to consolidate fear extinction
We postulated that fear extinction, being context-dependent[16], required SWRs as they play a crucial role in contextual memory consolidation through cortico-hippocampal circuits[22,23]. To test this, we suppressed SWRs by ventral hippocampal commissural electrical stimulation that induces phasic silencing of hippocampal pyramidal cells and interneurons[24–26]. Since animals trained with high-intensity footshocks tend to resist extinction, we reduced the training intensity (5 pairings CS+US at 0.7 mA) to ensure that the extinction criterion was achieved within seven sessions in control conditions. During stimulation following each extinction, online detected SWRs triggered a single-pulse (0.5 ms) ventral hippocampal commissural stimulation (Fig. 3a, b), with the stimulation intensity tailored to each animal's requirements to disrupt the SWRs (range: 5–15 V). Open-loop animals were randomly stimulated within the same voltage range.

The test results showed no significant differences in fear expression between groups after conditioning to CS+ (Fig. 3c) or during the first and last extinction day's initial 5 CS+ blocks, after conditioning to CS+ (Fig. 3d). However, animals that experienced SWR disruption required more extinction sessions to achieve an 80% reduction in freezing compared to those in the open-loop group (Fig. 3e). Additionally, these SWR-disrupted animals expressed elevated levels of freezing in the hybrid context during the renewal test compared to the

non-stimulated and open-loop groups (Fig. 3f). No differences were detected during the reinstatement test (Fig. 3g). These results suggest that hippocampal SWRs are essential for consolidating fear extinction. The disruption of SWRs results in slow extinction learning and fear persistence in different environments beyond the extinction context.

## The enhancement of extinction induced by closed-loop stimulation is mediated by D2 receptor and G protein Rac1 in BLA
We next explored the plasticity-dependent mechanisms that contribute to enhanced fear extinction induced by closed-loop MFB stimulation. We tested the potential involvement of BLA dopamine receptors and the small G protein Rac1, a Rho family member involved in learning-induced synapse formation[27–30]. After fear conditioning (5 pairings CS+US (1 mA)), animals received bilateral microinfusions of the Rac1 inhibitor NSC2376, D1R antagonist SCH23390, or D2R antagonist sulpiride immediately after each extinction session and before the closed-loop stimulation (Fig. 4a, b). The test results showed no significant differences after conditioning to CS+ (Fig. 4c), or in fear expression during the first 5 CS+ blocks from first and last extinction day (Fig. 4d).

Animals co-infused with NSC2376 and sulpiride required more days to achieve extinction than controls, closed-loop stimulated animals and closed-loop stimulated animals infused with SCH23390 (Fig. 4e). During the renewal test in the hybrid context, only sulpiride suppressed the effect of closed-loop stimulation (Fig. 4f).

Similar to the renewal test, animals infused with sulpiride exhibited a significant fear recovery after exposure to an immediate footshock protocol (Fig. 4g). The pharmacological treatments did not alter the extinction criterion without electrical stimulation. However, NSC2376 appeared to disrupt fear attenuation during renewal, suggesting that RAC1 itself participates in extinction consolidation (Supplementary Fig. 9). Thus, NSC2376 and sulpiride prevented the enhancement of extinction induced by the closed-loop neuromodulation. These findings suggest that closed-loop neuromodulation-induced fear extinction involves dendritic spine plasticity mediated by RAC1 signaling and D2Rs in the BLA.

## Discussion
Our study found that closed-loop stimulation of the MFB during SWRs was effective in enhancing the extinction of cued fear conditioning (Supplementary Fig. 10). We observed that stimulation without extinction learning or SWR-independent stimulation was ineffective. Our intervention resulted in a shortened time to reduce fear expression, and the effect persisted even 25 days after treatment, as animals were resistant to induced renewal, reinstatement, and spontaneous reemergence of fear expression.

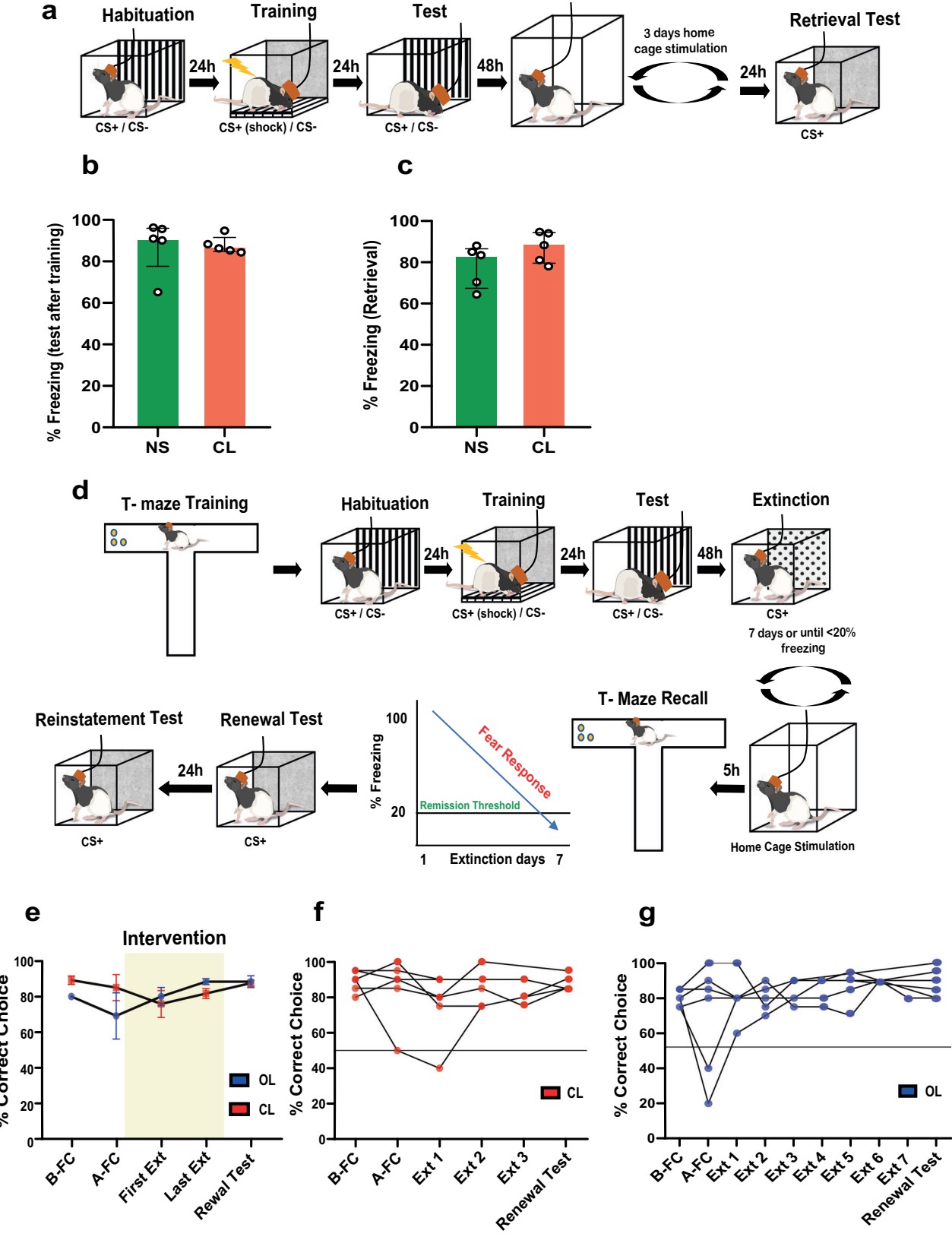

Our findings suggest that SWRs are essential for extinction learning, as disruption of SWRs increases the number of extinction sessions required for remission and predisposes animals to recurrent expression of fear (Supplementary Fig. 10). Closed-loop stimulation effects were mediated by D2 receptors and RAC1 signaling in the BLA, suggesting that closed-loop modulation of the reward pathways promotes a plasticity-dependent mechanism leading to extinction. These results offer novel avenues to develop closed-loop neuromodulation technologies for PTSD and anxiety disorders.

Conventional deep brain stimulation (DBS) introduces preset electrical stimulation in an open-loop manner, without being aligned to the internal oscillatory activity. Although DBS has been used to

**Fig. 2 | Contribution of fear extinction and side effects on co-storaged memories during closed-loop MFB stimulation. a** Schematics of the experimental design. Fear conditioning and test was performed as before. Closed-loop animals were exposed to 3 consecutive SWR-triggered stimulation sessions without extinction. No difference was found in fear expression in response to the CS+ following training (Mann–Whitney test: $U = 7$, $P = 0.3095$, two-tailed) **b** and renewal (Mann–Whitney test: $U = 6$, $P = 0.2222$, two-tailed) **c** between the groups (non-stimulated (NS) $n = 5$; closed-loop (CL) $n = 5$). **d** Before fear conditioning, animals were trained in a visual cue forced alternation T-maze task until achieving 80% of correct

choice. Next, animals were exposed to fear conditioning, extinction and stimulation following Fig. 1. **e** T-maze performance was unaltered during the experiments regardless of the stimulation type (Unpaired $t$ test: $P > 0.05$ on all instances, two-tailed) (open-loop (OL) $n = 6$; closed-loop (CL) $n = 6$). Individual performance of the animals is shown for open-loop **f** and closed-loop **g**. Bar plots and error bars represent medians and interquartile ranges, individual data points are also displayed. Detailed statistics are shown in Supplementary Data 1. Source data provided as a Source Data file. Silhouettes on **a** and **d** are obtained from https://github.com/eackermann/ratpack under MIT License.

control fear expression in animal models[31,32] and humans[33], open-loop approaches may be excessive and disrupt normal physiological oscillations[34]. Closed-loop stimulation may reduce frequent side effects such as strabismus during MFB-DBS stimulation reported by patients with major depression[35].

SWRs encode and consolidate spatial memory and are involved in fear memory processing. Selective pre or post-training inactivation of CA3 disrupts the acquisition and consolidation of contextual fear memory by reducing the number and dominant frequency of CA1 ripples and shifting underlying CA1 ensemble activity[36]. SWRs rely on synchronous CA1 principal neuron activation mainly controlled by PV+ interneurons[37]. Boosting the activity of hippocampal PV+ interneurons results in selective extinction of contextual fear memory and increased SWR incidence[38]. However, suppression of hippocampal PV+ interneurons alters principal neuronal phase coupling to SWRs, decreasing ripple-spindle coupling and consolidation of contextual fear memory[39,40]. Our findings indicate that SWRs are necessary for the extinction of cued fear conditioning and can update the memory trace with rewarding information. Closed-loop disruption of SWRs delayed but did not block extinction since 80% of animals still achieved the remission criterion, consistent with the contextual dependence of fear extinction[16,18,19], although cued fear conditioning is amygdala-dependent[41–43]. Our initial hypothesis that SWRs encode contextual features of 'safety' during the extinction is supported by the decreased time to achieve remission but does not explain the fear reduction to CS+.

SWRs play a critical role in establishing temporally precise 'windows' for integrating information across neocortical and subcortical structures. A widespread increase in neocortical activity precedes SWRs[44] indicating that during SWRs, replay, and information integration involve the contextual features of an engram and the corresponding emotional memory traces. The multiple roles of SWRs and hippocampal place cells in processing contingencies beyond spatial localization support this idea[45,46]. Thus, the SWR-triggered closed-loop MFB stimulation and the resulting reward signal coincide with the widespread ongoing brain network activity orchestrating the consolidation of fear extinction[47] during SWR events. Neuronal activity in the BLA increases during SWRs[48,49] and coordinated reactivation between the dorsal hippocampus and BLA during offline aversive memory processing peaks around the SWRs[50]. Therefore, the SWR-triggered closed-loop neuromodulation may provide a reward safety signal to a consolidating aversive memory[51] and/or enhance the network activity that encodes fear extinction[52]. Since SWRs are also important in encoding context, it cannot be excluded that the enhancement shown in this study might also influence spatial or contextual learning. While we demonstrated that the closed-loop SWR-triggered MFB stimulation does not interfere with already consolidated spatial memories, revealing any effects on their acquisition or extinction may require further studies.

The potential mechanism underlying the closed-loop neuromodulation of SWRs and reward signaling resembles a counter-conditioning process by memory updating with contrasting emotional valence[53–56] characterized by high temporal and neurochemical precision. This hypothesis is supported by the absence of closed-loop effect when animals are not exposed to the extinction learning. In such cases,

the reward signal triggered by MFB stimulation does not coincide with extinction-contingent SWRs, which prevent the enhancement of fear attenuation.

Interestingly, when the US is unexpectedly omitted during extinction, there is an increase in the activity of dopaminergic neurons in the VTA[57]. This increase in activity has a positive correlation with extinction learning. In addition, optogenetic excitation of VTA dopaminergic neurons at the time of the US omission accelerates fear extinction[58]. These results suggest that dopaminergic activity during extinction encodes prediction error or mismatch between expectancies[59]. This system is more active during the initial phase (unexpected omission) compared to late phase (expected omission) of extinction learning[58]. Together our results from MFB closed-loop neuromodulation, combined with the assumption that US omission may be rewarding itself[60], suggest that dopaminergic signaling plays a crucial role in the consolidation of extinction during offline states, particularly during SWRs.

The idea that dopaminergic signaling is essential for extinction consolidation is supported by the fact that MFB fibers, which connect nodes involved in reward and emotional processing, play a critical role in this process. The VTA sends dopaminergic axons to the NAc, amygdala and PFC via the MFB[61]. A cluster of dopaminergic neurons in the anterior VTA/SNc directly connect with CA1[62]. A global manipulation of the reward system through MFB deep brain stimulation can ameliorate depression-like behaviors in animal models and depression symptoms in human patients[63]. We found that temporally precise electrical stimulation in these circuits during SWRs may scaffold the extinction enhancement. We argue for a dopamine-dependent mechanism, since previous studies have shown that MFB stimulation leads to an increase in dopamine release in BLA[64–66] and the effects of closed-loop neuromodulation were prevented by a selective local antagonism of D2 but not D1 receptors. Moreover, our stimulation protocol was able to induce conditioned place preference.

Multiple lines of evidence supports that fear conditioning induces long-term potentiation of amygdala principal neurons[67] and fear extinction can revert the enhanced activity of these neurons and decrease AMPAR expression induced by fear conditioning[68]. Dopamine enhances the excitability of BLA projection neurons, and D1 and D2 receptor activation increase excitability and input resistance, respectively[69].

Indeed, dopamine release in the BLA during fear learning is controlling the saliency of the footshock and the extinction through prediction error signaling of non-reinforced CS+ presentation[70]. Fear memories and extinction are encoded by different BLA neuronal populations. Rather than overwriting the original fear learning engrams, extinction engrams can suppress the activity of neurons that were initially engaged in fear learning. Furthermore, since neurons that mediate extinction learning also overlap with those involved in reward processing, the activation of these neurons could also signal reward[20].

Our experimental design cannot differentiate whether post-extinction SWRs are related to the reactivation of the original fear memory or represent the consolidation of the extinction. However, increased dopamine release during SWRs could change the emotional valence of an engram replay or directly suppress neurons engaged in

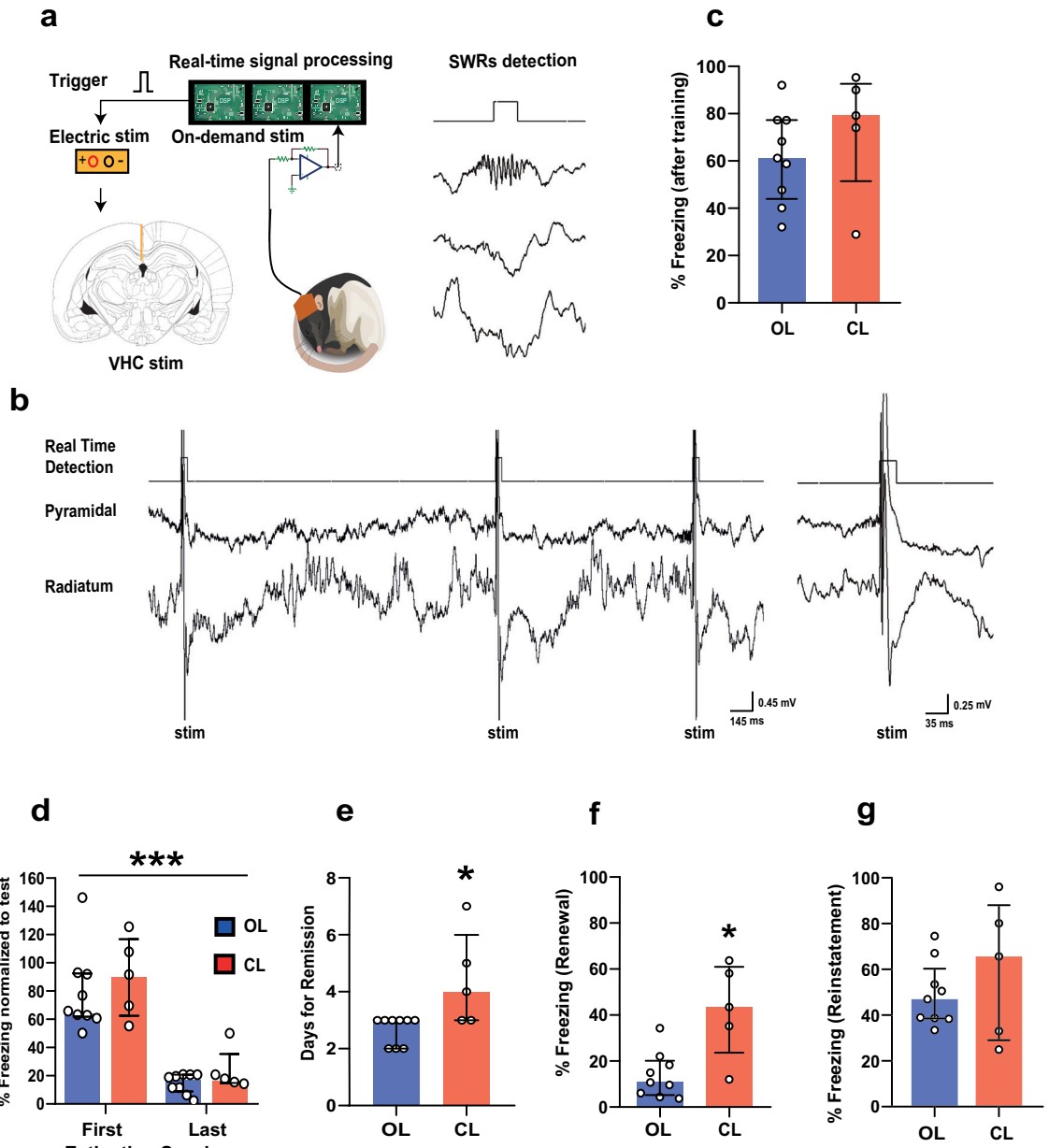

**Fig. 3 | SWRs are required for the extinction of fear memories. a** The behavioral protocol was performed as before, but SWR-triggered VHC stimulation was performed for 1 h following each extinction session. **b** Representative LFP signals from dorsal hippocampus showing intact and disrupted SWR events. **c** No difference in fear expression in response to the CS+ following training between the two experimental groups (Mann–Whitney test: $U = 14$, $P = 0.2977$, two-tailed; open-loop (OL) $n = 9$; closed-loop (CL) $n = 5$). **d** No difference between the fear expression of the two experimental groups during the first 5 CS+ block from first and last extinction day. However, there was a significant decrease in fear expression over time (mixed ANOVA: $F (1,12) = 65.00$, $P < 0.0001$, time factor) **e** SWR-disrupted animals require more days to achieve the extinction criterion (Mann–Whitney test: $U = 6$, $P = 0.0280$, two-tailed). **f** SWR-disrupted animals show high fear expression during renewal (Mann–Whitney test: $U = 4$, $P = 0.0120$, two-tailed). **g** No difference in fear expression during reinstatement (Mann–Whitney test: $U = 20$, $P = 0.7972$, two-tailed). *$P < 0.05$, ***$P < 0.001$. The bar plots and error bars represent medians and interquartile ranges, and individual data points are also displayed. Detailed statistics are shown in Supplementary Data 1. Source data provided as a Source Data file. Silhouettes on **a** are obtained from https://github.com/eackermann/ratpack under MIT License.

fear learning. Reward-responsive VTA neuronal activity is coupled to SWRs during quiet wakefulness[71], supporting the idea that dopamine release is modulated by SWRs. Dopaminergic projections from VTA innervate PV+ interneurons expressing D2 receptors, contributing to the suppression of BLA principal neurons[72]. The suppression of feed-forward inhibition can induce LTP at excitatory afferent synapses in the BLA, an effect also mediated by D2 receptors[73]. Although the initial fear generalization phenotype was not evaluated after our closed-loop intervention, there is evidence that cue fear generalization is

promoted by high-intensity training[74] and is a limiting factor for extinction[75]. Generalization may be mediated by the temporal proximity between CS+ and CS-, linking memory traces by neuronal co-allocation to overlapping engrams[76]. Given this scenario, it is expected that closed-loop MFB stimulation would impact not only the CS+ trace but also the overlapping CS-trace. This hypothesis should be addressed in future studies.

Dopamine stimulation of engram cells may enhance forgetting by activating Rac1/Cofilin, which modulates actin cytoskeleton and

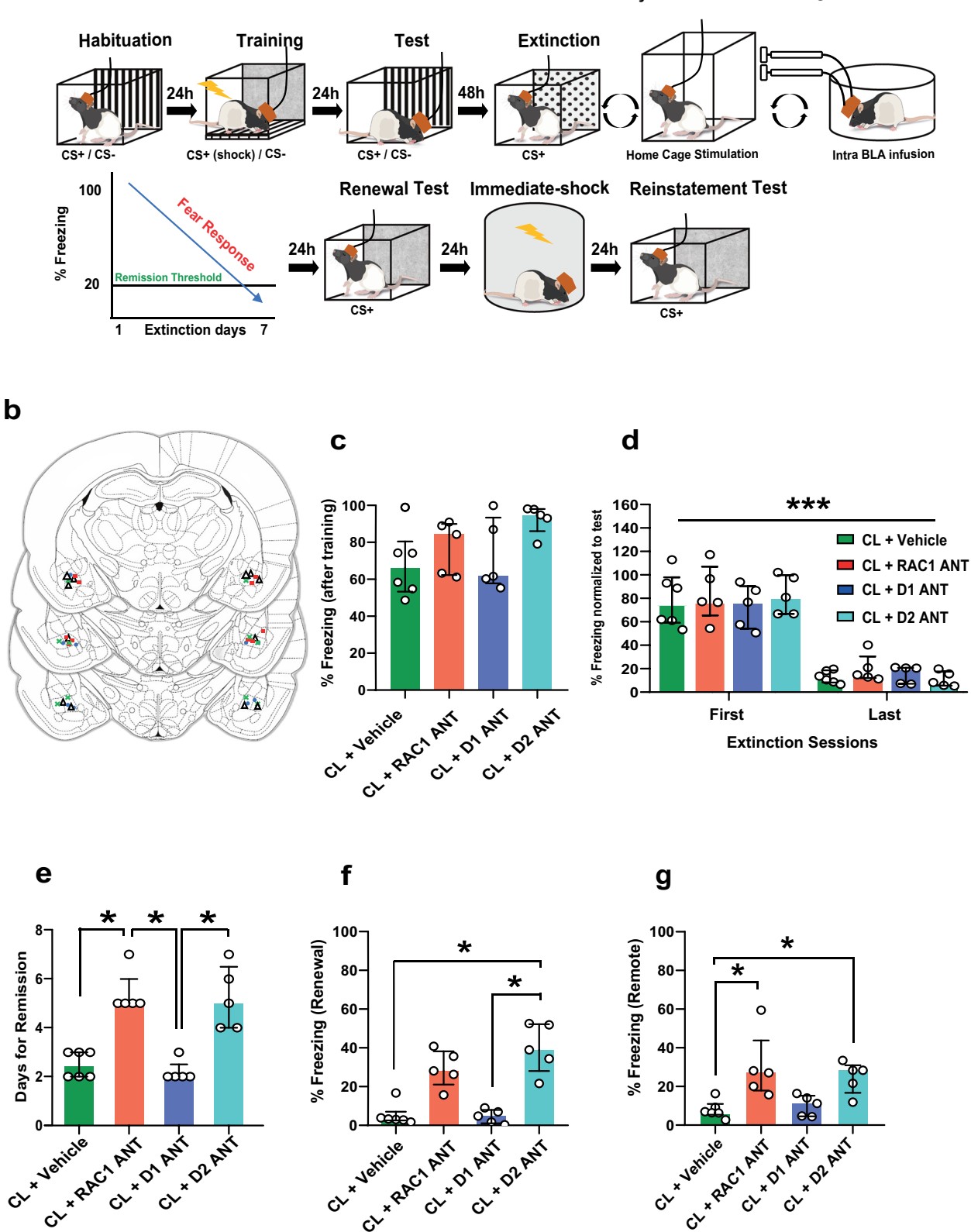

cellular morphology[2]. Inhibition of Rac1 activity in the dHPC impairs extinction of contextual fear memories[77] and photoactivation of Rac1 in the motor cortex suppresses motor learning[29].

Our findings suggest three sequential mechanisms underpinning closed-loop extinction enhancement: (1) SWRs reactivate the memory trace in BLA. (2) Closed-loop MFB stimulation promotes concurrent dopamine release in BLA. (3) BLA dopamine release can induce D2 receptor-mediated plasticity processes culminating in Rac1 activation. Blocking Rac1 signaling prevents spontaneous or closed-loop neuro-modulation-induced fear reduction during renewal. However, Rac1 inhibition without closed-loop neuromodulation did not extend the number of sessions required for successful fear extinction using the

**Fig. 4 | The closed-loop neuromodulation-induced enhancement of extinction is mediated by Rac1 and D2Rs in the BLA. a** The behavioral protocol and closed-loop neuromodulation were performed as before and immediately after each extinction session, the BLA was bilaterally microinfused with the Rac1 inhibitor NSC2376, D1R antagonist SCH23390, or D2R antagonist sulpiride. **b** The locations of the cannula tips in each animal are shown, with colors representing the different experimental groups. **c** No significant difference in fear expression was observed in response to the CS+ following training between the four experimental groups (Kruskal–Wallis test: $H = 5.430$, $P = 0.1429$) (closed loop (CL + Vehicle) $n = 6$; closed-loop + NSC2376 (CL + RAC1 ANT) $n = 5$; closed-loop + SCH23390 (CL + D1 ANT) $n = 5$; closed-loop + sulpiride (CL + D2 ANT) $n = 5$). **d** No difference between the fear expression of the four experimental groups during the first 5 CS+ block from first

and last extinction day. However, there was a significant decrease in fear expression over time (mixed ANOVA: $F (1,34) = 175.1$, $P < 0.0001$, time factor). **e** NSC2376 and sulpiride injected animals required more extinction sessions to achieve the extinction criterion (Kruskal–Wallis test: $H = 16.16$, $P = 0.0011$). **f** Sulpiride suppress the extinction enhancement induced by closed-loop neuromodulation during renewal (Kruskal–Wallis test: $H = 14.84$, $P = 0.0020$). **g** Animals treated with NSC2376 and sulpiride exhibited fear recovery compared to animals injected with vehicle (Kruskal–Wallis test: $H = 12.55$, $P = 0.0057$). Bar plots and error bars represent medians and interquartile ranges, individual data points are also displayed. Detailed statistics are shown in Supplementary Data 1. Source data provided as a Source Data file. Silhouettes on **a** and **b** are obtained from https://github.com/eackermann/ratpack under MIT License.

remission criterion. This partial disruption without electrical stimulation is expected since previous studies have shown that RAC1 inhibition impairs the extinction of contextual fear memories[77] and RAC1 activation is required for plasticity-related mechanism during fear extinction[78]. Since the disruption in fear attenuation was more pronounced under closed-loop stimulation, the synaptic plasticity may differ between normal and enhanced extinction. Additional work is required to determine the mechanisms of interaction between dopamine receptors and Rac1 modulation during fear extinction.

Our results suggest a novel translational treatment of fear-related disorders. The US Food and Drug Administration (FDA) approved MFB stimulation for treatment-resistant depression in clinical trials, with promising efficacy[63,79]. It should be noted that in addition to the MFB, other components of the reward system, as the nucleus accumbens (NAc), may also be viable targets for closed-loop neuromodulation. This is supported by evidence demonstrating that NAc-DBS can elicit striatal dopamine release in humans[80]. Although in our experiments the detection of SWRs was invasive, alternatively, cortical slow-waves and spindles concurring with SWRs in animals[81,82] may be detected non-invasively to align stimulation. Thus, closed-loop stimulation triggered by cortical EEG activity could replace SWRs detection. Further, non-invasive techniques (e.g., tDCS, TMS) could stimulate reward-associated cortical areas instead of penetrating electrodes. Importantly, our experiments were performed in male animals only. Considering sex differences in the renewal and the context-dependence of extinction in rodents[83] as well as the higher risk of women to develop anxiety-related disorder compared to men[84,85], future experiments are needed to assess if the closed-loop neuromodulation approach can be extended to females.

Our framework to study and attenuate fear-related memories relies on closed-loop stimulation guided by classical biomarkers of memory consolidation. Closed-loop stimulation can reduce the side effects from chronic and excessive stimulation of DBS approaches. Temporally precise manipulation of the reward system during SWRs overcomes the resistance to extinction in an animal model with key features of PTSD. Moreover, SWRs are critical for extinction learning. Although dopaminergic agonists can enhance fear extinction[86,87], our intervention avoids the side effects of systemic treatments (e.g., psychosis, pathological gambling). Coupling between SWRs, cortical slow-waves, and spindles may offer a potential way to translate our approach towards a non-invasive therapy in the future.

## Methods
### Animals
Rats (120 adult male Long-Evans, 300–450 g, 3–6 months old) were kept in a 12-hour light/ dark cycle. All experiments were performed in accordance with the European Union guidelines (2003/65/CE) and the National Institutes of Health Guidelines for the Care and Use of Animals for Experimental Procedures. The experimental protocols were approved by the Ethical Committee for Animal Research at the Albert Szent-Györgyi Medical and Pharmaceutical Center of the University of Szeged (XIV/218/2016 and XIV/824/2021).

### Surgery
The animals were anesthetized with 2% isoflurane and craniotomies were performed according to stereotaxic coordinates. Intracortical electrode triplets (interwire spacing, 0.2–0.4 mm)[88] targeting the anterior cingulate cortex (ACC) (AP: +1.0, ML: 0.5, DV: 1.4), bilateral BLA (AP:−2.8, ML: 4.6, DV: 8.1 mm from the dura) and the bilateral CA1 subfield of the dorsal hippocampus (AP: −3.5, −4.5, and −5.5, ML: 2.0, 3.0 and 4.0, DV: 2.9 and 3.0 all mm from Bregma). To improve DH-SWRs detection, a custom-built microdrive[89] was used in some experiments, allowing the vertical adjustment over the CA1 subfield. A custom-built bipolar electrode consisting of two insulated (except 200 μm at the tip) Tungsten wires (interwire spacing, 0.4 mm) was implanted in the left medial-forebrain bundle (AP: −2.8, ML: 2.0 mm, DV: 8.1 all mm from Bregma). LFP electrodes and the base of the microdrive were secured to the skull with dental acrylic (Unifast Trad, USA). Two stainless-steel screws above the cerebellum served as ground and reference for the recordings, respectively. A Faraday cage was built using copper mesh and dental acrylic on the skull around the implanted electrodes.

In experiments involving concomitant electrophysiological recording and local pharmacological infusion, in addition to electrodes, rats were bilaterally implanted with 25-gauge guide cannulas above the BLA (AP: −2.8, ML: 4.7, DV: 6.9 all mm from Bregma). Cannulae were fixed to the skull with dental acrylic (Unifast Trad). Caps were used to cover cannulae to avoid any accidental occlusion.

### Electrophysiological recordings and stimulation
Rats were housed individually in Plexiglass home cages (42 × 38 cm, 18 cm tall). LFP recordings were performed in the home cage and the fear conditioning box (see below). Recording and stimulation sessions for closed-loop or open-loop interventions were performed during the first hour following the extinction protocol. To avoid any twisting and over-tension of the recording cables, a bore-through electrical commutator (VSR-TC-15-12; Victory-Way Electronic) was used. Food and water were available *ad libitum*. All recording sessions took place in the same room using 12/12 h light/dark cycle with light onset/offset at 7 h/ 19 h The multiplexed signals were acquired at 500 Hz per channel for closed-loop neuromodulation experiments[88]. The neuronal signals were preamplified (total gain 400×), multiplexed on head, and stored after digitalization at 20 kHz sampling rate per channel (KJE1001, Amplipex, Szeged, Hungary). During home cage stimulation, pre-amplified signals were analyzed online by a programmable digital signal processor (RX-8, Tucker-Davis Technologies, Alachua, FL, USA) using a custom-made sharp-wave ripple detection algorithm, as follows.

Two LFP signals were used for real-time SWRs detection. For ripple detection, a channel from the tripolar electrodes from CA1 pyramidal layer with the largest ripple amplitude was selected and band-pass filtered (150–250 Hz), and root-mean-square (RMS) power was calculated in real-time for ripple detection. For noise detection, manual inspection from channels of the ACC, VHC, or AMY was performed to select the signal with no ripple-like activity and lower noise

incidence to enhance signal-to-noise ratio during detection. In case of the ACC, signal was filtered between 80 and 500 Hz. SWRs were defined as events crossing the ripple thresholds in the absence of the noise signal. Amplitude threshold for ripple was adjusted for each animal before fear conditioning training. SWRs were defined as events crossing ripple thresholds in the absence of the noise signal in the neocortical site. Threshold crossings triggered a stimulation train lasting 100 ms and composed of fourteen 1-ms long, 100 μA square-wave pulses at 140 Hz) in the MFB or single pulse (5–15 V in the ventral hippocampal commissure (VHC) (STG4008; Multi Channel Systems, Reutlingen, Germany) depending on the experiment performed. MFB stimulation was performed under current mode and VHC stimulation in voltage-controlled mode. The threshold of the detection algorithm was set for each rat separately. Behavioral (i.e., rewarding) effect of MFB stimulation was confirmed with a place preference task (see below).

### Electrophysiological data analysis

The offline ripples were analyzed using custom-made MATLAB (R2017b, Natick, Massachusetts, USA) routines. Raw signals were down sampled from 20 kHz to 500 Hz and bandpass filtered in the ripple band (150–250 Hz) of hippocampal channels. Normalized squared signal was calculated. Putative SWRs events were defined as those where the beginning/end cutoffs exceeded 2 SDs and the peak power 3 SDs. The detection window was set in 150 ms. SWR duration limits were set to be between 20 and 200 ms, otherwise the events were excluded to minimize artifacts. All ripple events were drawn out for manually speculations after offline detection. The closest stimulation onset from the digital channel was selected for further analysis. Then calculated the time delay between the successfully detected ripples events and the stimulation time. For the brain states classifications (SWS/REM), SleepScoreMaster toolbox from Buzcode (https://github.com/buzsakilab/buzcode) was employed combined with post manually corrections. Time-frequency spectrum was calculated in MATLAB using Multitaper Spectral Estimation from the Chronux Toolbox (http://chronux.org/). A 2 s sliding window with a 50% overlap, a time-bandwidth product of 5 and tapers of 3 were chosen.

### Drugs and infusions

The Rac1 inhibitor NSC2376 (10 μg/μl), D1 dopamine receptor antagonist SCH23390 (0.50 μg/μl), and D2 dopamine receptor antagonist sulpiride (1 μg/μl) were dissolved in sterile physiological saline (0.9% NaCl). NSC2376, SCH23390, and sulpiride were infused bilaterally into the BLA using a 33 G gauge injectors connected to Hamilton syringes via 20-gauge plastic tubes. The infusion injectors tip protruding 2.0 mm below the tip of the cannula and aimed the BLA center. A total volume of 0.5 μl per side was infused by a microinfusion pump at a rate of 0.125 μl/min. Injectors were left in place for an additional minute to ensure proper drug diffusion. All drugs were infused after the extinction sessions.

### Auditory fear conditioning

The experiments were carried out in a fear conditioning apparatus comprising three contextual Plexiglas boxes (42 × 38 cm, 18 cm tall) placed within a soundproof chamber. Four different contextual configurations were used (Habituation and Test Context (A): square configuration, white walls with black vertical horizontal lines, white smooth floor, washed with 70% ethanol; Training Context (B): square configuration, gray walls, metal grid on black floor, washed 30% ethanol; Extinction Context (C): rectangular configuration, white walls with black dots, white smooth floor; and Renewal and Remote/Reinstatement context (D): hybrid context comprising a square configuration, gray walls from training context, white smooth floor, washed with 70% ethanol. All sessions were controlled using a MATLAB custom script.

### Habituation.
On day 1, animals were exposed to the habituation session in context A. After 2 min of contextual habituation, they were exposed to 5 alternating presentations of two different tones (2.5 or 7.5 kHz, 85 dB, 30 s). Tone time intervals were randomized (30–40 s) during the session. No behavioral differences were detected under exposition to the two frequencies.

### Training.
On day 2, cue fear conditioning was performed in context B. After 2 min of contextual habituation, animals received 5 trials of one tone (CS+: 7.5 kHz) immediately followed by a 2 s long footshock as unconditioned stimulus (US: 1.0 mA, 0.7 mA or 0.5 mA, depending on the experiment performed). The other tone (CS−: 2.5 kHz) was presented 5 times intermittently but never followed by the US.

### Test.
On day 3, animals underwent fear retrieval in context A. After 2 min of contextual habituation, rats were exposed to presentations of the CS+ or CS- in two different sessions. Each session consisted of a block of five tones. The order of the CS+ and the CS− in each session was randomized. Sessions were repeated every 4–6 h.

### Extinction.
In context C, from day 5 until reaching the remission criterion (see below), rats received extinction training consisting of twenty CS+ presentations without the US (unreinforced tones). Tones were repeated with randomized intervals (30–40 s) during the session.

### Fear remission from extinction.
We used an extinction threshold criterion to assess the efficacy of fear reduction after extinction sessions similar to[90]. The block of the first five tones during each extinction session was assessed to determine fear reduction level of the given day. Considering individual differences under fear conditioning[90–92] fear reduction during extinction was expressed as a fraction of the percentage of freezing expressed during the CS+ test (Day 3) (% Freezing Reduction = Freezing extinction × 100/Freezing test CS+). Fear remission was considered achieved when animals expressed <20% of the initial freezing during the first block of the day (i.e., first 5 CS+ presentations during the extinction session). Extinction training was repeated for maximum 7 days.

### Renewal and remote test.
Twenty-four hours or 25 days after achieving the remission, animals were exposed to context D (Hybrid context) as a renewal or remote test, respectively. In each test, rats were exposed to a block of five CS+ presentations after 2 min of contextual habituation. Time intervals between tones were randomized (30–40 s) during the session.

### Immediate footshock.
To promote fear recovery, animals were placed in a neutral environment outside the conditioning box and received an unconditioned footshock after 30 s contextual exposition, with the same intensity used during fear conditioning. The animals were returned to their home cage 30 s following the footshock.

### Reinstatement test.
Animals were submitted to a reinstatement test in context D 24 hours after the immediate footshock. Rats were exposed to a block of 5 CS+ presentations after 2 min of contextual habituation. Time intervals between tones were randomized during the session.

### Behavioral assessment.
Freezing behavior was used as a memory index in the fear conditioning task. Freezing was analyzed offline using Solomon software (SOLOMON CODER, © András Péter, Budapest, Hungary), for behavioral coding by an experienced observer that was blinded to the experimental group. Freezing was defined as the absence of all movements, except those related to breathing, while the animal was alert and awake.

## Conditioned place preference

The conditioning box consisted of three chambers, two for the conditioning session having the same dimensions (24 × 40 × 50 cm), and the other serving as a central/start chamber (10 × 40 × 50 cm). Each chamber was employed with contextual cues and floor texture to distinguish them.

Conditioned place preference test consisted of three phases: pre-conditioning (day 1), conditioning (days 2–6), and test (day 7). The pre-conditioning session (15-min) was intended to reduce novelty and determine initial preferences for any of the two chambers by assessing the time spent in each compartment. Conditioning always took place in the initially less preferred chamber. Conditioning sessions were performed during the following five days. Animals underwent two conditioning sessions each day with 6–8 h intervals between sessions. In one session, animals were placed in the initially less preferred compartment and received MFB stimulation (duration: 20 min, same intensity as used during fear conditioning experiments). During the other session, the animals were placed in the opposite compartment without stimulation. The order of the sessions was randomized between animals and days. A 15 min place preference test was conducted in the absence of stimulation 24 h after the last conditioning day. The video of the animal behavior was recorded and analyzed offline using the ANY-Maze (Stoelting, Wood Dale, IL, USA, Version 7.20) video tracking software.

## T-maze task

Animals on food restriction (no less than 85% of their baseline weight) were habituated to the T-maze during 5 days before the training. The T-maze was constructed from black acrylic, with 80 cm long and 30 cm wide alleys and 40 cm high walls. Two removable doors closed the side alleys. During training, a light cue indicated the correct arm to receive a reward (froot-loops pellet). A total of 20 trials per day were performed until achieving 80% of correct choice. A removable door in the central arm was used to confine the animal at the starting point during cue presentation. After 3 min, the alley was removed, and the animal allowed to run in the maze. After arm selection, the alley was closed and the animal remains additional 3 min in the maze before next trial. Afterwards, fear conditioning, extinction, and stimulation sessions started. Animals were tested in the T-maze after the extinction sessions to verify any disruption of the consolidated spatial memory. Extinction and stimulation sessions and T-maze tests were separated by five hours and the order of the behavioral tasks were randomized each day.

## Histology

Following the termination of the experiments, animals were deeply anesthetized with 1.5 g/kg urethane (i.p.), and the recording sites of each electrode were lesioned with 100 μA anodal direct current for 10 s (Supplemental Fig. 1C). Then, the animals were transcardially perfused with 0.9% saline solution followed by 4% paraformaldehyde solution and 0.2% picric acid in 0.1 M phosphate buffer saline. After postfixation overnight, 50 μm thick coronal sections were prepared with a microtome (VT1000S, Leica), stained with 1 μg/ml DAPI in distilled water (D8417; Sigma-Aldrich), coverslipped, and examined using a Zeiss LSM880 scanning confocal microscope (Carl Zeiss) and the software ZEN Digital Imaging for Light Microscopy (RRID: SCR_013672) for histological verification of the recording electrode and cannulae locations (Fig. 4b and Supplementary Fig. 7).

## Statistical analysis

Statistical analyses were performed using GraphPad Prism 8 software. Significance was set at $p < 0.05$. Data were analyzed using two-tailed Mann–Whitney $U$ test, Kruskal–Wallis test, or Mixed ANOVA followed by Dunn's post hoc or Bonferroni's multiple comparisons test. Data are expressed and visualized as median ± IQR, individual data points are also shown where applicable. Detailed statistics are shown in Supplementary Data 1.

## Reporting summary

Further information on research design is available in the Nature Portfolio Reporting Summary linked to this article.

## Data availability

The data generated in this study (in the main manuscript and in the Supplementary Information) are provided in the Source Data file and Supplementary Data 1, or from the corresponding author upon request. Source data are provided with this paper.

## Code availability

All custom code is freely available from the corresponding author on request.

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

## Acknowledgements

We thank Laura Herrera and Johanna Duran for their technical assistance. This work was supported by the Momentum program II of the Hungarian Academy of Sciences (A.B.), EFOP-3.6.1-16-2016-00008 (A.B.), EFOP-3.6.6-VEKOP-16-2017-00009 (A.B.), and KKP133871/KKP20 grants of the National Research, Development and Innovation Office, Hungary (A.B.), the 20391-3/2018/FEKUSTRAT of the Ministry of Human Capacities, Hungary (A.B.), and the EU Horizon 2020 Research and Innovation Program (No. 739593—HCEMM to A.B.), Ministry of Innovation and Technology of Hungary grant TKP2021-EGA-28 (A.B.), Hungarian Scientific Research Fund (Grants NN125601 and FK123831 to M.L.L.), the Hungarian Brain Research Program (grant KTIA_NAP_13-2-2014-0014 (M.L.L.) and NAP2022-I-7/2022 (A.B. & M.L.L.)), UNKP-20-5 New National Excellence Program of the Ministry for Innovation and Technology from the source of the National Research, Development and Innovation Fund (M.L.L.), Premium Postdoctoral Research Program of the Hungarian Academy of Sciences (ROS), Japan Agency for Medical Research and Development grant 22zf0127004h0002 and 22gm6510015h0001 (Y.T). M.L.L. was a grantee of the János Bolyai Fellowship.

## Author contributions

R.O.S., L.K.P., and A.B. conceived the project. R.O.S., L.K.P., G.K., A.J.N., Y.T., and A.B. developed the methodology. R.O.S., L.K.P., L.B., Q.L., and A.P. performed the experiments and analyzed data. R.O.S., L.K.P., M.L.L., O.D., G.B., and A.B. wrote the manuscript. O.D., G.B. advised the project. A.B. supervised the project.

## Funding

## Competing interests

A.B. is the owner of Amplipex Llc. Szeged, Hungary a manufacturer of signal-multiplexed neuronal amplifiers. A.B. is a shareholder, chairman,

and CEO, O.D. is an advisor and director, and G.B. is a shareholder of Neunos Inc, a Boston, MA company, developing neurostimulator devices. The remaining authors declare no competing interests.

## Additional information

**Peer review information** *Nature Communications* thanks Stephen Maren and the other, anonymous, reviewer(s) for their contribution 2 to the peer review of this work. A peer review file is available.

