## [Peer Review File · Nature Communications]

Closed-loop brain stimulation augments fear extinction in male ratsEditorial Note: This manuscript has been previously reviewed at another journal that is not operating a transparent peer review scheme. This document only contains reviewer comments and rebuttal letters for versions considered at *Nature Communications*.

REVIEWERS' COMMENTS

Reviewer #1 (Remarks to the Author):

The authors have addressed most of my points. I nevertheless still have some comments.

The authors mentioned that they have tone down their claims about fear conditioning being a model of PTSD. However as early as in the abstract (line 38-39) they mentioned otherwise. As already mentioned, if the authors claim is that they are modelling PTSD they should demonstrate the face, predictive and construct validity of their model.

I am bit confused by the answer of the authors on the requirement for additional control and spatial learning experiments. Our first request was to demonstrate that extinction of the spatial task associated with MFB stimulation has no effect on spatial learning compared to controls: this control is still missing if I am correct. This is important especially because the authors were not able to evaluate the effect of MFB stimulation during extinction on the learning of a new spatial task.

Otherwise, very nice piece of work

Reviewer #2 (Remarks to the Author):

The reviewers have prepared a very responsive revision and have satisfied my concerns. This will be a timely and important paper for the scientific community.

Reviewer #3 (Remarks to the Author):

The manuscript is improved, and most of my concerns have been addressed. However, the introduction is not as clear as it should be and, in my opinion, needs a major rewrite. Firstly, it is not sufficiently focused. The logic of the experiments is quite simple, and the authors would do better to stick to the central theme-- can rewarding brain stimulation augment extinction learning? Secondly, along with other reviewers, I do not think that fear conditioning is an adequate model of PTSD, and the over-emphasis on PTSD in the introduction feels like an overreach, and I recommend sticking to describing how understanding the basic mechanisms underlying fear learning and extinction may be relevant to treatment of fear/anxiety disorders.

Reviewers' Comments:

Reviewer #1 (Remarks to the Author):

The authors have addressed most of my points. I nevertheless still have some comments.

The authors mentioned that they have tone down their claims about fear conditioning being a model of PTSD. However as early as in the abstract (line 38-39) they mentioned otherwise. As already mentioned, if the authors claim is that they are modelling PTSD they should demonstrate the face, predictive and construct validity of their model.

We have substantially rewritten the abstract and introduction to have less emphasis on PTSD.

I am bit confused by the answer of the authors on the requirement for additional control and spatial learning experiments. Our first request was to demonstrate that extinction of the spatial task associated with MFB stimulation has no effect on spatial learning compared to controls: this control is still missing if I am correct. This is important especially because the authors were not able to evaluate the effect of MFB stimulation during extinction on the learning of a new spatial task.

In our manuscript we show that our manipulations do not interfere with the established spatial memories. Whether the same manipulation can interfere with spatial learning and extinction learning remains to be established by future studies. We now discuss this topic in the manuscript's discussion (page: 11, lines: 253-257):

"Since SWRs are also important in encoding context, it cannot be excluded that the enhancement shown in this study might also influence spatial or contextual learning. While we demonstrated that the closed-loop SWR-triggered MFB stimulation does not interfere with already consolidated spatial memories, revealing any effects on their acquisition or extinction may require further studies."

Otherwise, very nice piece of work

We thank the Reviewer for her/his appreciative words.

Reviewer #2 (Remarks to the Author):

The reviewers have prepared a very responsive revision and have satisfied my concerns. This will be a timely and important paper for the scientific community.

We thank the Reviewer for her/his appreciative words.

Reviewer #3 (Remarks to the Author):

The manuscript is improved, and most of my concerns have been addressed. However, the introduction is not as clear as it should be and, in my opinion, needs a major rewrite. Firstly, it is not sufficiently focused. the logic of the experiments is quite simple, and the authors would do better to

stick to the central theme-- can rewarding brain stimulation augment extinction learning? Secondly, along with other reviewers, I do not think that fear conditioning is an adequate model of PTSD, and the over-emphasis on PTSD in the introduction feels like an overreach, and I recommend sticking to describing how understanding the basic mechanisms underlying fear learning and extinction may be relevant to treatment of fear/anxiety disorders.

We agree with the Reviewer and have substantially rewritten the introduction to have less emphasis on PTSD and structured the logic in answering the question “can rewarding brain stimulation augment extinction learning?”.

Please note that the basolateral amygdala participates in both negatively reinforced and rewarded behaviors, facilitated via a previously evidenced process of mutual inhibition (Kim, J et al., 2016). Specifically, neurons responsible for extinction exhibit overlap with those responsive to natural reward (Zhang, et al., 2020). This information has been incorporated into the discussion (page: 13, lines: 285-295) and is also mentioned in the introduction of our manuscript, where it is linked to our primary hypothesis (page: 3-4, lines: 63-68):

“Excitatory neurons in the basolateral amygdala have been shown to respond to both reward and punishment and have been proposed to be involved in mediating reward signaling induced by the omission of an unconditioned stimulus during extinction²⁰. Additionally, these neurons participate in a mutual inhibition process²¹. Based on these findings, we hypothesize that manipulating internal reward signals during extinction learning could facilitate the extinction of memories, thereby reducing excessive fear reactions in inappropriate contexts.”

References

- Kim, J., Pignatelli, M., Xu, S., Itohara, S. & Tonegawa, S. Antagonistic negative and positive neurons of the basolateral amygdala. *Nat Neurosci* 19, 1636-1646 (2016).
- Zhang, X., Kim, J. & Tonegawa, S. Amygdala Reward Neurons Form and Store Fear Extinction Memory. *Neuron* 105, 1077-1093 e1077 (2020).